# SepVAE: a contrastive VAE to separate pathological patterns from healthy ones

**R. Louiset**[1,2]**, E. Duchesnay**[2]**, A. Grigis**[2]**, B. Dufumier**[1,2]**, P. Gori**[1]

[1] *LTCI, Télécom Paris, IPParis, France*

[2] *NeuroSpin, CEA, University Paris-Saclay, France*

## Abstract

Contrastive Analysis VAE (CA-VAEs) is a family of Variational auto-encoders (VAEs) that aims at separating the common factors of variation between a *background* dataset (BG) (*i.e.,* healthy subjects) and a *target* dataset (TG) (*i.e.,* patients) from the ones that only exist in the target dataset. To do so, these methods separate the latent space into a set of **salient** features (*i.e.,* proper to the target dataset) and a set of **common** features (*i.e.,* exist in both datasets). Currently, all CA-VAEs models fail to prevent sharing of information between the latent spaces and to capture all salient factors of variation. To this end, we introduce two crucial regularization losses: a disentangling term between common and salient representations and a classification term between background and target samples in the salient space. We show a better performance than previous CA-VAEs methods on three medical applications and a natural images dataset (CelebA). [1]

**Keywords:** Contrastive Analysis, VAE, generative model, Psychiatry, population analysis.

## 1. Introduction

One of the goals of unsupervised learning is to learn a compact, latent representation of a dataset, capturing the underlying factors of variation. Furthermore, the estimated latent dimensions should describe distinct, noticeable, and semantically meaningful variations. One way to achieve that is to use a generative model, like Variational Auto-Encoders (VAEs) (Kingma and Welling, 2013), (Higgins et al., 2017) and disentangling methods (Higgins et al., 2017), (Burgess et al., 2018), (Shu et al., 2018), (Ainsworth et al., 2018), (Li et al., 2018). Differently from these methods, which use a *single* dataset, in Contrastive Analysis (CA), researchers attempt to distinguish the latent factors that generate a *target* (TG) and a *background* (BG) dataset. Usually, it is assumed that target samples comprise additional (or modified) patterns with respect to background data. The goal is thus to estimate the **common** generative factors and the ones that are **target-specific** (or **salient**).

For instance, consider two sets of data: 1) healthy neuro-anatomical MRIs (BG=*background dataset*) and 2) Alzheimer-affected patients' MRIs (TG=*target dataset*). As in (Jack, 2018; Antelmi et al., 2019; Dufumier et al., 2021), given these two datasets, neuroscientists would be interested in distinguishing common factors of variations (*e.g.:* effects of aging, education or gender) from Alzheimer's specific markers (*e.g.:* temporal lobe atrophy, an increase of beta-amyloid plaques). Until recently, separating the various latent mechanisms that drive neuro-anatomical variability in neuro-degenerative disorders was considered hardly feasible. This can be attributed to the intertwining between the variability due to natural

---

1. Code and datasets available at `https://github.com/neurospin-projects/2023_rlouiset_sepvae`.

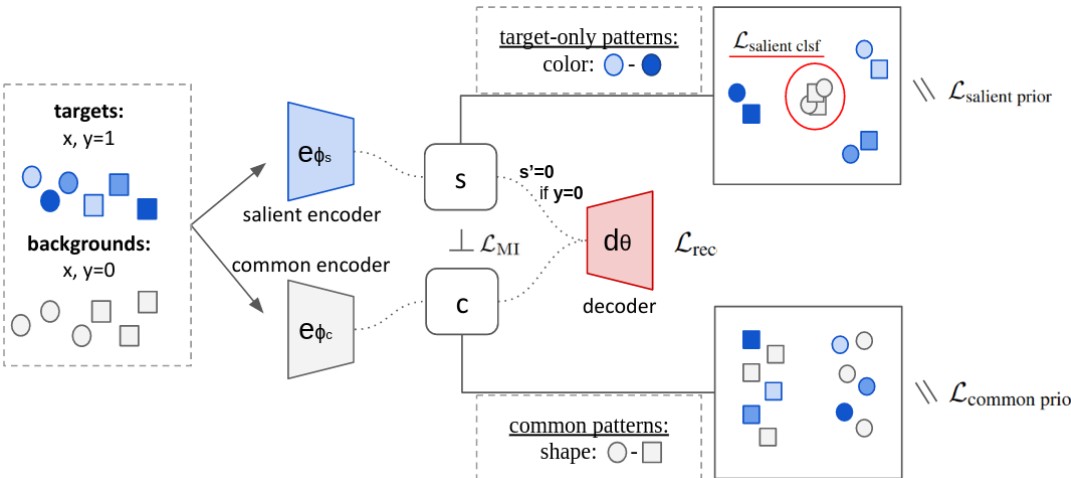

Figure 1: Illustration of SepVAE training. Target ($y = 1$) and background ($y = 0$) images are encoded with the same encoders $e_{\phi_s}$ and $e_{\phi_c}$. The first encoder $e_{\phi_s}$ estimates the salient factors of variation $s$ of the target samples. Background samples' salient space is set to an informationless value $s' = 0$. The second encoder $e_{\phi_c}$ estimates the common factors $c$. Images are reconstructed using a single decoder $d_\theta$ fed with the concatenation of $\mathbf{c}$ and $\mathbf{s}$. The common space $\mathbf{c}$ should only capture common factors of variability (shape), while the salient space $\mathbf{s}$ should model target-only factors of variability (color).

aging and the variability due to neurodegenerative disease development. The combined effects of both processes make hardly interpretable the discovery of novel bio-markers. The objective of developing such a Contrastive Analysis method would be to help separate these processes. And thus identifying correlations between neuro-biological markers and pathological symptoms. In the **common features** space, aging patterns should correlate with normal cognitive decline, while **salient features** (*i.e.:* Alzheimer-specific patterns) should correlate with pathological cognitive decline.

## 2. Related works

Variational Auto-Encoders (VAEs) (Kingma and Welling, 2013) have advanced the field of unsupervised learning by generating new samples and capturing the underlying structure of the data onto a lower-dimensional data manifold. Disentangling methods (Higgins et al., 2017; Burgess et al., 2018; Shu et al., 2018) enable learning the underlying factors of variation in the data. While disentangling (Zheng and Sun, 2019; Chen et al., 2019) is a desirable property for improving the control of the image generation process and the interpretation of the latent space (Ainsworth et al., 2018; Li et al., 2018), these methods are usually based on a *single* dataset, and they do not explicitly use labels or multiple datasets to effectively estimate and separate the common and salient factors of variation.
Contrastive VAE (Abid and Zou, 2019; Weinberger et al., 2022; Severson et al., 2019; Ruiz et al., 2019; Zou et al., 2022; Choudhuri et al., 2019) have employed deep encoders in order to capture higher-level semantics. They usually rely on a latent space split into two parts, a common and a salient, produced by two different encoders. First methods, such as (Severson

et al., 2019), employed two decoders (common and salient) and directly sum the common and salient reconstructions in the input space. This seems to be a very strong assumption, probably wrong when working with high-dimensional and complex images. For this reason, subsequent works used a single decoder, which takes as input the concatenation of both latent spaces. Importantly, when seeking to reconstruct background inputs, the decoder is fed with the concatenation of the common part and an informationless reference vector **s'**. This is usually chosen to be a null vector in order to reconstruct a null (i.e., empty) image by setting the decoder's biases to 0. To fully enforce the constraints and assumptions of the underlying CA generative model, previous methods have proposed different regularizations. Here, we analyze the most important ones with their advantages and shortcomings:

**Minimizing background's variance in the salient space** Pioneer works (Abid and Zou, 2019) have shown inconsistency between the encoding and the decoding task. While background samples are reconstructed from **s'**, the salient encoder does not encourage the background salient latents to be equal to **s'**. To fix that, posterior works (Weinberger et al., 2022; Zou et al., 2022; Choudhuri et al., 2019) proposed to explicitly nullifying the background variance in the salient space. This regularization is necessary to avoid salient features explaining the background variability but not sufficient to prevent information leakage between common and salient spaces, as shown in (Weinberger et al., 2022).

**Independence between common and salient spaces** Only (Abid and Zou, 2019) proposed to prevent information leakage between the common and salient space by minimizing the total correlation (TC) between $p(c, s|x)$ and $p(c|x) \times p(s|x)$. Similarly to FactorVAE (Kim and Mnih, 2019), they used the density-ratio trick (Nguyen et al., 2010), which requires to *independently* train a discriminator $D_\lambda(.)$ to approximate the ratio between $p(c, s|x)$ and $p(c|x) \times p(s|x)$. However, (Abid and Zou, 2019)'s code does *not* use an independent optimizer for $\lambda$, which is theoretically wrong, and it thus undermines their contribution.

**Matching background and target common patterns** Another work (Weinberger et al., 2022), has proposed to encourage the distribution in the common space to be the same across target samples and background samples. In practice, we argue that it may encourage undesirable *biases* to be captured by salient factors rather than common factors. For example, suppose that we have healthy subjects (*background* dataset) and patients (*target* dataset) and that patients are composed of both young and old individuals, whereas healthy subjects are mostly old (*i.e.,* imbalance dataset). We would expect the CA method to capture the normal aging patterns in the common space. However, forcing both $p(c|x, y = 0)$ and $p(c|x, y = 1)$ to follow the same distribution in the common space would probably bring to a biased distribution and thus to leakage of information between salient and common factors (i.e., aging could be considered as a salient factor of the patient dataset).This behavior is not desirable, and we believe that the statistical independence between common and salient space is a more robust property. Our contributions are three-fold:

• We develop a new Contrastive Analysis method, called SepVAE, which is supported by a sound and versatile Evidence Lower BOund maximization framework.

• We identify and implement two properties: the salient space discriminability and the salient/common independence, that have not been successfully addressed by previous Contrastive VAE methods.

• We provide a fair comparison with other SOTA CA-VAE methods on 3 medical applications and a natural image experiment.

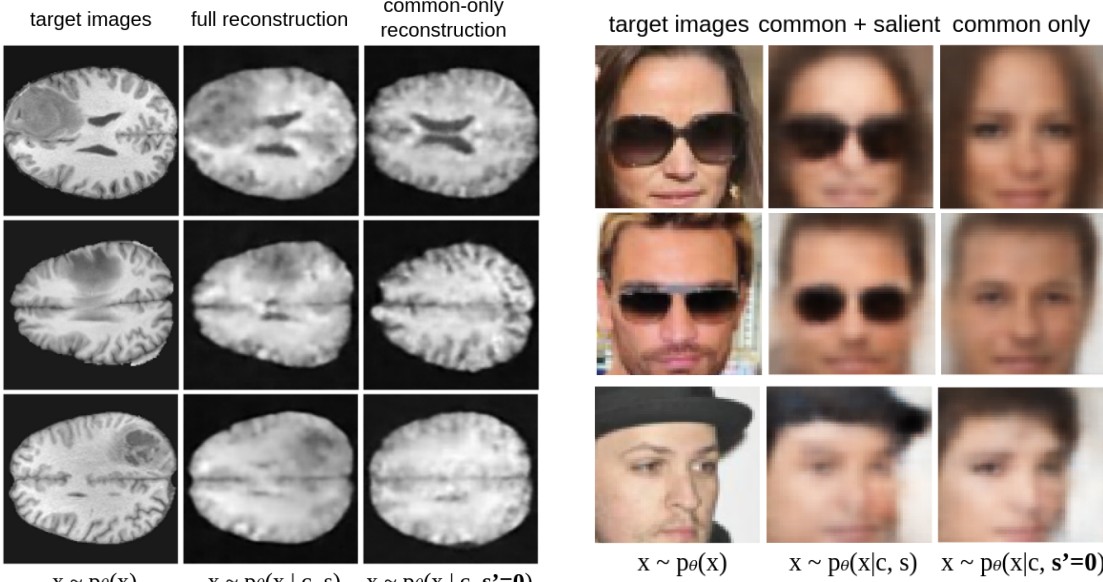

Figure 2: SepVAE. Reconstructions on BRATS dataset ([Menze et al., 2015](#)), we separate healthy patterns from tumors.

Figure 3: SepVAE. Reconstructions with CelebA accessories dataset (BG = no accessories, TG = hats and glasses).

## 3. Contrastive Variational Autoencoders

Let $(X, Y) = \{(x_i, y_i)\}_{i=1}^N$ be a data-set of images $x_i$ associated with labels $y_i \in \{0, 1\}$, 0 for background and 1 for target. Both background and target samples are assumed to be i.i.d. from two different and unknown distributions that depend on two latent variables: $c_i \in \mathbf{R}^{D_c}$ and $s_i \in \mathbf{R}^{D_s}$. Our objective is to have a generative model $x_i \sim p_\theta(x|y_i, c_i, s_i)$ so that: 1- the **common** latent vectors $C = \{c_i\}_{i=1}^N$ should capture the common generative factors of variation between the background and target distributions and fully encode the background samples and 2- the **salient** latent vectors $S = \{s_i\}_{i=1}^N$ should capture the distinct generative factors of variation of the target set (*i.e.,* patterns that are only present in the target dataset and not in the background dataset). Similarly to previous works([Abid and Zou, 2019](#); [Weinberger et al., 2022](#); [Zou et al., 2022](#)), we assume the generative process: $p_\theta(x, y, c, s) = p_\theta(x|c, s, y)p_\theta(c)p_\theta(s|y)p(y)$. Since $p_\theta(c, s|x, y)$ is hard to compute in practice, we approximate it using an auxiliary parametric distribution $q_\phi(c, s|x, y)$ and directly derive the Evidence Lower Bound of $\log p(x, y)$:

$$- \log p_\theta(x, y) \le \mathbf{E}_{c, s \sim q_{\phi_c, \phi_s}(c, s|x, y)} \log \frac{q_{\phi_c, \phi_s}(c, s|x, y)}{p_\theta(x, y, c, s)} \qquad (1)$$

Then, we can develop the lower bound into three terms, a conditional reconstruction term, a common space prior regularization, and a salient space prior regularization. From there, we assume the independence of the auxiliary distributions (*i.e.:* $q_{\phi_c, \phi_s}(c, s|x, y) = q_{\phi_c}(c|x)q_{\phi_s}(s|x, y)$) and prior distributions (*i.e.:* $p_\theta(c, s) = p_\theta(c)p_\theta(s)$). Both $p_\theta(x|y_i, c_i, s_i)$ (i.e., single decoder) and $q_{\phi_c}(c|x)q_{\phi_s}(s|x, y)$ (i.e., two encoders) are assumed to follow

a Gaussian distribution parametrized by a neural network. To reinforce the independence assumption between $c$ and $s$, we introduce a Mutual Information regularization term $KL(q(c,s)||q(c)q(s))$. This property is desirable in order to ensure that the information is well separated between the latent spaces. Theoretically, this term is similar to the one in (Abid and Zou, 2019). However, in (Abid and Zou, 2019), the Mutual Information estimation and minimization are done simultaneously [2], which is theoretically wrong (see Sec. 2). Here, we correctly implement an independent optimizer to estimate the Mutual Information. To further reduce the overlap of target and common distributions on the salient space, differently from previous works, we also introduce a salient classification loss defined as $\mathbf{E}_{s\sim q_{\phi_s}(s|x,y)} \log p(y|s)$. By combining all these losses together, we obtain the final loss $\mathcal{L}$:

$$\mathcal{L} = \underbrace{-\mathbf{E}_{c,s\sim q_{\phi_c,\phi_s}(c,s|x,y)} \log p_\theta(x|c,s,y)}_{\textbf{a) Conditional Reconstruction}} + \underbrace{KL(q(c,s)||q(c)q(s))}_{\textbf{e) Mutual Information}} - \underbrace{\mathbf{E}_{s\sim q_{\phi_s}(s|x,y)} \log p_\theta(y|s)}_{\textbf{d) Salient Classification}}$$
$$+ \underbrace{KL(q_{\phi_c}(c|x)||p_\theta(c))}_{\textbf{b) Common Prior}} + \underbrace{KL(q_{\phi_s}(s|x,y)||p_\theta(s|y))}_{\textbf{c) Salient Prior}}$$

$$(2)$$

**Conditional reconstruction** The reconstruction term is $-\mathbf{E}_{c,s\sim q_{\phi_c,\phi_s}(c,s|x,y)} \log p_\theta(x|c,s,y)$. Given an image $x$ (and a label $y$), a common and a salient latent vector can be drawn from $q_{\phi_c,\phi_s}$ with the help of the reparameterization trick. We assume that $p(x|c,s,y) \sim \mathcal{N}(d_\theta([c,ys+(1-y)s'],I)$, *i.e:* $p_\theta(x|c,s,y)$ follows a Gaussian distribution parameterized by $\theta$, centered on $\mu_{\hat{x}} = d_\theta([c,ys+(1-y)s'])$ with identity covariance matrix, and $d_\theta$ is the decoder and $[.,.]$ denotes a concatenation. Therefore, by developing the reconstruction loss term, we obtain the mean squared error between the input and the reconstruction: $\mathcal{L}_{\text{rec}} = \sum_{i=1}^{N} ||x - d_\theta([c,ys+(1-y)s'])||_2^2$. Importantly, as in (Weinberger et al., 2022; Abid and Zou, 2019), we set the salient latent vectors of background samples to **s'** $= 0$. This choice enables isolating the background factors of variability in the common space only.

**Common prior** Assuming $p(c) \sim \mathcal{N}(0,I)$ and $q_{\phi_c}(c|x) \sim \mathcal{N}(\mu_\phi(x),\sigma_\phi(x,y))$, the KL loss has a closed form solution, as in usual VAE. Here, both $\mu_\phi(x)$ and $\sigma_\phi(x,y)$ are the outputs of the encoder $e_{\phi_c}$. This loss is also used in (Abid and Zou, 2019; Weinberger et al., 2022).

**Salient prior** First, we develop $p_\theta(s) = \sum_y p(y)p_\theta(s|y)$, where $p(y)$ follows a Bernoulli distribution with probability equal to 0.5. This allows us to distinguish the salient priors of background samples ($p(s|y=0)$) and target samples ($p(s|y=1)$). Similar to other CA-VAE methods, we assume that $p(s|y=1) \sim \mathcal{N}(0,I)$ and , as in (Zou et al., 2022), that $p(s|x,y=0) \sim \mathcal{N}(s',\sqrt{\sigma_p}I)$, with $s' = 0$ and $\sqrt{\sigma_p} < 1$, namely a Gaussian distribution centered on an informationless reference $s'$ with a small constant variance $\sigma_p$. We preferred it to a Delta function $\delta(s = s')$ (as in (Weinberger et al., 2022)) because it eases the computation of the KL divergence (i.e., closed form) and it also means that we tolerate a small salient variation (e.g., noisy/erroneous diagnosis labels) in the background samples.

**Salient classification** The salient prior regularization encourages BG and TG salient factors to match two different Gaussian distributions centered in $s' = 0$, but with different covariance. To further reduce the overlap of target and common distributions on the salient

---

2. In (Abid and Zou, 2019), Alg. 1 suggests that the MI estimation and minimization depend on two distinct parameter updates. However, in their code, a single optimizer is used. Moreover, in Sec. 3, authors write: "discriminator is trained simultaneously with the encoder and decoder".

space, we propose to minimize a Binary Cross Entropy (BCE) loss to distinguish the target from background samples in the salient space. Assuming that $p(y|s)$ follows a Bernoulli distribution parameterized by $f_\xi(s)$, a 2-layers classification Neural Network, we obtain a BCE loss between true labels $y$ and predicted labels $\hat{y} = f_\xi(s)$. This loss is *not* used in (Abid and Zou, 2019; Weinberger et al., 2022).

**Mutual Information** To promote independence between $c$ and $s$, we minimize their mutual information, defined as the KL divergence between the joint distribution $q(c, s)$ and the product of their marginals $q(c)q(s)$. However, computing this quantity is not trivial, and it requires a few tricks to correctly estimate and minimize it. As in (Abid and Zou, 2019), it is possible to take inspiration from FactorVAE (Kim and Mnih, 2019), which proposes to estimate the density-ratio between a joint distribution and the product of the marginals. In our case, we seek to enforce the independence between two sets of latent variables rather than between each latent variable of a set. The density-ratio trick (Nguyen et al., 2010; Sugiyama et al., 2012) allows us to estimate the quantity inside the log in Eq.3. First, we sample from $q(c, s)$ by randomly choosing a batch of images $(x_i, y_i)$ and drawing their latent factors $[c_i, s_i]$ from the encoders $e_{\phi_c}$ and $e_{\phi_s}$. Then, we sample from $q(c)q(s)$ by using the same batch of images where we shuffle the latent codes among images (*e.g.*, $[c_1, s_2]$, $[c_2, s_3]$, etc.). Once we obtained samples from both distributions, we trained an **independent** classifier $D_\lambda([c, s])$ to discriminate the samples drawn from the two distributions by minimizing a BCE loss. The classifier is then used to approximate the ratio in the KL divergence, and we can train the encoders $e_{\phi_c}$ and $e_{\phi_s}$ to minimize the resulting loss:

$$\mathcal{L}_{\text{MI}} = \mathbb{E}_{q(c,s)} \log \left( \frac{q(c, s)}{q(c)q(s)} \right) \approx \sum_i \text{ReLU} \left( \log \left( \frac{D_\lambda([c_i, s_i])}{1 - D_\lambda([c_i, s_i])} \right) \right) \tag{3}$$

## 4. Experiments

**Evaluation** We evaluate the ability of SepVAE to separate common from target-specific patterns on three medical and one natural (CelebA) imaging datasets. We compare it with the only SOTA CA-VAE methods whose code is available: MM-cVAE (Weinberger et al., 2022) and ConVAE [3] (Abid and Zou, 2019), using the same architecture for all models. For quantitative evaluation, we use the fact that the information about some attributes (e.g. glasses/hats in CelebA) should be present either in the common or in the salient space. Once the encoders/decoder are trained, we train a Logistic (or Linear) Regression on the estimated salient and common factors of the training set to predict the attribute presence (or value). Then, we evaluate the classification/regression model on the salient and common factors estimated from a test set. We also report the background (BG) vs target (TG) classification accuracy (Acc.) using the trained classifier for SepVAE and an independently trained classifier (still 2 layers MLPs) for the other methods.

**CelebA - glasses vs hat identification:** In the CelebA with attributes dataset (Liu et al.), the target set contains images of celebrities wearing glasses or hats while background images show no accessories. We used a train set of 20000 images, (10000 no accessories, 5000 glasses, 5000 hats) and an independent test set of 4000 images (2000 no accessories, 1000 glasses, 1000 hats). In Tab.1 and Fig. 3, we demonstrate that we successfully distinguish

---

3. ConVAE implemented with our MI minimization, *i.e.:* with independently trained discriminator.

Table 1: CA-VAE performance on CelebA with accessories dataset. Accessories (glasses/hat) information should only be present in the salient space, not in the common. Average and std are computed over 5 different runs with different parameter initialization.

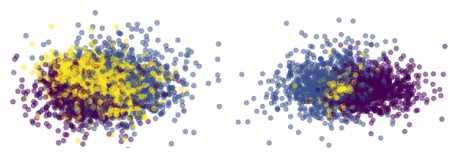

Figure 4: PCA projections of MM-cVAE (left) and SepVAE (right) salient space on CelebA TEST set. Yellow: no accessories. Dark Blue: glasses. Purple: hats.

|  | Glss/Hats Acc salient ↑ | Glss/Hats Acc common ↓ | Bg vs Tg AUC salient ↑ | Bg vs Tg AUC common ↓ |
|---|---|---|---|---|
| ConVAE | 82.32±1.17 | 75.01±2.52 | 82.46±0.58 | 78.39±0.41 |
| MM-cVAE | 85.17±0.60 | 73.93±1.66 | 88.53±0.39 | 78.03±0.35 |
| SepVAE | **87.62±0.75** | **72.16±2.02** | **93.15±1.65** | **77.60±0.20** |

glasses and hats attributes in the salient space[4]. In Fig. 4, we show that SepVAE, differently from MM-cVAE, maximizes the target variance in the salient space while reducing the background variance. Ratios of variances are: MM-cVAE: $\sigma^2(s|y=0)/\sigma^2(s|y=1]) = 1.79$; SepVAE: $\sigma^2(s|y=0])/\sigma^2(s|y=1) = 20.31$. More details are in the Supplementary.

Table 2: CA-VAE methods performance on the Healthy vs Pneumonia X-Ray dataset. Pneumonia subtype information should only be present in the salient space. The lower part shows an ablation study of regularization losses. Average and std are computed over 5 different runs with different parameters initializations.

|  | Subgrp Acc salient ↑ | Subgrp Acc common ↓ | Bg vs Tg Acc salient ↑ | Bg vs Tg Acc common ↓ |
|---|---|---|---|---|
| ConVAE := SepVAE no SAL + CLSF | 82.30±1.53 | 73.58±1.84 | 67.80±5.93 | 58.05±7.17 |
| MM-cVAE | 82.86±1.87 | 74.35±3.19 | 70.44±2.69 | 59.94±5.88 |
| SepVAE | **84.78±0.42** | **70.92±1.39** | **78.13±3.03** | 57.52±4.14 |
| SepVAE no **MI** | 84.10±0.48 | 71.792±2.94 | 75.186±5.69 | 60.35±4.73 |
| SepVAE no **CLSF** | 84.71±1.19 | 73.58±2.19 | 71.91±4.65 | **55.79±5.41** |
| SepVAE no **SAL** | 83.98±0.85 | 72.61±2.05 | 73.03±2.97 | 61.43±2.25 |
| SepVAE no **MI + SAL** | 81.58±3.68 | 71.73±5.17 | 61.24±3.89 | 54.33±5.30 |
| SepVAE no **MI + CLSF** | 84.25±0.47 | 73.17±3.15 | 53.10±1.63 | 57.58±6.74 |
| SepVAE no **MI + SAL + CLSF** | 81.78±2.12 | 76.71±2.10 | 62.87±7.15 | 59.37±5.69 |

**Pneumonia subgroups:** From (et al., 2018), we used 1342 healthy radiographies (*background*) and 2684 pneumonia radiographies (*target*), divided into two subgroups: viral (1342 samples) and bacterial (1342 samples), see Fig.7 in the Suppl. In Tab. 2, we demonstrate that our method can produce a salient space that captures the pathological variability, as it better distinguishes the two subgroups.

Ablation: In Tab. 2, we also propose to disable different components of the loss to show that the proposed full model is always better on average. no **MI** means that we removed the Mutual Information loss. no **CLSF** means that we disabled the Salient Classification loss. no **SAL** means that we ignored the Salient Prior loss.

**Parsing neuro-anatomical variability in psychiatric diseases** Given a background population of Healthy Controls (HC) and a target population suffering from a Mental Disorder (MD), the objective is to capture the pathological factors of variability in the salient space, such as psychiatric and cognitive clinical scores, while isolating in the common space the patterns related to demographic variables, such as age and sex, or acquisition sites. For each experiment, we gather T1w anatomical VBM (Ashburner and Friston, 2000) preprocessed images of HC and MD subjects. We divide them into 5 TRAIN, VAL splits (0.75, 0.25) and evaluate in a cross-validation scheme the performance of SOTA CA-VAEs.

**Schizophrenia:** We merged images of schizophrenic patients (TG) and healthy controls (BG) from the datasets SCHIZCONNECT-VIP (Wang et al., 2016) and BSNIP (Tamminga

---

4. Our evaluation process is different from (Weinberger et al., 2022) as their TEST set has been used during the model training. Here, TRAIN and TEST are correctly separated.

Table 3: CA-VAE methods performance on the prediction of disorder-specific variables, *i.e.* SANS, SAPS, for schizophrenia disorder (upper table), and ADI-s, ADOS (Akshoomoff et al., 2006), for autism disorder (lower table) and common variables (Age, Sex, Site) using only salient factors of test images from the target dataset. MAE=Mean Absolute Error.

| | Age MAE ↑ | Sex B-Acc ↓ | Site B-Acc ↓ | SANS MAE ↓ | SAPS MAE ↓ | Diag AUC ↑ |
|---|---|---|---|---|---|---|
| ConVAE | 7.46±0.18 | 72.72±1.32 | 54.46±2.46 | **3.95±0.28** | 2.76±0.18 | 58.53±4.87 |
| MM-cVAE | 7.10±0.34 | **72.15±2.47** | 56.69±9.84 | 4.52±0.33 | 3.16±0.05 | 70.94±4.08 |
| SepVAE | **7.98±0.25** | 72.61±2.19 | **44.10±5.78** | 4.14±0.39 | **2.60±0.27** | **79.15±3.39** |

| | Age MAE ↑ | Sex B-Acc ↓ | Site B-Acc ↓ | ADOS MAE ↓ | ADI-s MAE ↓ | Diag AUC ↑ |
|---|---|---|---|---|---|---|
| ConVAE | 3.97±0.19 | 66.67±1.12 | 40.97±2.06 | 10.1±1.27 | 5.14±0.17 | 54.93±2.04 |
| MM-cVAE | 3.74±0.12 | 64.07±2.58 | 40.93±2.66 | 10.5±2.47 | 5.09±0.16 | 54.88±2.76 |
| SepVAE | **4.38±0.09** | **59.61±1.78** | **33.58±1.86** | **8.55±1.68** | **4.91±0.17** | **59.73±1.78** |

et al., 2014). Results in Tab. 3 show that the salient factors estimated using our method better predict schizophrenia-specific variables of interest: SAPS (Scale of Positive Symptoms), SANS (Scale of Negative Symptoms), and diagnosis. On the other hand, salient features are shown to be poorly predictive of demographic variables: age, sex, and acquisition site. It paves the way toward a better understanding of schizophrenia disorder by capturing neuro-anatomical patterns that are predictive of the psychiatric scales while not being biased by confound variables (Barbano et al., 2023). More details in the Suppl.

**Autism:** We also combine patients with autism from ABIDE1 and ABIDE2 (Heinsfeld et al., 2017) (TG) with healthy controls (BG). In Tab. 3, SepVAE's salient latents better predict the diagnosis and the clinical variables, such as ADOS (Autism Diagnosis Observation Schedule) and ADI Social (Autism Diagnosis Interview Social) which quantifies the social interaction abilities. On the other hand, salient latents poorly infer irrelevant demographic variables (age, sex, and acquisition site). More details in the Suppl.

## 5. Conclusions and Perspectives

Building onto Contrastive Analysis methods, we discuss previously proposed regularizations about (1) the matching of target and background distributions in the common space and (2) the overlapping of target and background priors in the salient space. These regularizations may fail to prevent information leakage between common and salient spaces. We thus propose two alternative solutions: salient discrimination between target and background samples, and mutual information minimization between common and salient spaces. We demonstrate superior performances on radiological and two neuro-psychiatric applications, where we successfully separate the pathological information of interest (diagnosis, pathological scores) from the "nuisance" common variations (e.g., age, site). The development of CA methods offers a large spectrum of perspectives. It could be further extended to multiple target datasets (healthy Vs several pathologies) and to other generative models, such as GANs or Diffusion Models, for improved generation quality. Furthermore, generative models could also be coupled with contrastive learning losses, to improve the representation quality (Dufumier et al., 2023).

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

## Appendix A. Context on Variational Auto-Encoders

Variational Autoencoders (VAEs) are a type of generative model that can be used to learn a compact, continuous latent representation of a dataset. They are based on the idea of using an encoder network to map input data points $x$ (*e.g:* an image) to a latent space $z$, and a decoder network to map points in the latent space back to the original data space. Mathematically, given a dataset $X = x_{i_{i=1}^N}$ and a VAE model with encoder $q_\phi(z|x)$ and decoder $p_\theta(x|z)$, the VAE seeks $\phi, \theta$ to maximize a lower bound of the input distribution likelihood:

$$\log p_\theta(x) \leq \mathbf{E}_{z \sim q_\phi(z|x)} \log p_\theta(x|z) - KL(q_\phi(z|x)||p_\theta(z)) \tag{4}$$

where $p_\theta(x|z)$ is the likelihood of the input space, and $\mathrm{KL}(q_\phi(z|x)||p(z))$ is the Kullback-Leibler divergence between $q_\phi(z|x)$, the approximation of the posterior distribution, and $p(z)$ the prior over the latent space (often chosen to be a standard normal distribution). The first term in the objective function, $\mathbf{E}_{z \sim q_\phi(z|x)} \log p_\theta(x|z)$, is the negative reconstruction error, which measures how well the decoder can reconstruct the input data from the latent representation. The second term, $\mathrm{KL}(q_\phi(z|x)||p(z))$, encourages the encoder distribution to be similar to the prior distribution, which helps to prevent overfitting and encourage the learned latent representation to be continuous and smooth.

## Appendix B. Salient posterior sampling for background samples

In Sec. 3.3, we motivated the choice of a peaked Gaussian prior for salient background distribution with a user-defined $\sigma_p$. This way, the derivation of the Kullback-Leiber divergence is directly analytically tractable as in standard VAEs.

To simplify the optimization scheme, we could also set and freeze the standard deviations $\sigma_q^{y=0}$ of the salient space of the background samples. This way, it reduces the Kullback-Leiber divergence between $q_\phi(s|x, y=0)$ and $p_\theta(s|x, y=0)$ to a $\frac{1}{\sigma_p}$-weighted Mean Squared Error between $\mu_s(x|y=0)$ and $s'$ : $\frac{||\mu_s^{x_i|y=0} - s'||_2^2}{\sigma_p}$.

In our code, we make this choice as it simplifies the training scheme ($\sigma_q^{y=0}$ does not need to be estimated). In the case where there exists a continuum between healthy and diseased populations, $\sigma_q^{y=0}$ should be estimated.

Also, the choice of a frozen $\sigma_q^{y=0}$ allows controlling the radius of the classification boundary between background and target samples in the salient space. Indeed, the classifier is fed with samples from the target distributions ($q_{\phi_s(s|x,y=1)} \sim N(\mu_s(x), \sigma_s(x))$), and background distributions ($q_{\phi_s(s|x,y=0)} \sim N(\mu_s(x|y=0), \sigma_q)$). This implicitly avoids the overlap of both distributions with a margin proportional to $\sigma_q$. See Fig. 5 for a visual explanation.

In real applications, in particular medical ones, diagnosis labels can be noisy, and mild pathological patterns may exist in some healthy control subjects. Using such a prior, we tolerate these possible (erroneous) sources of variation. Furthermore, one could also extend the proposed method to a continuous $y$, for instance, between 0 and 1, describing the severity of the disease. Indeed, practitioners could define a function $\sigma_p(y)$ that would map the severity score $y$ to a salient prior standard deviation (*e.g.,* $\sigma_p(y) = y$). In this way, we could extend our framework to the case where pathological variations would follow a continuum from no (or mild) to severe patterns.

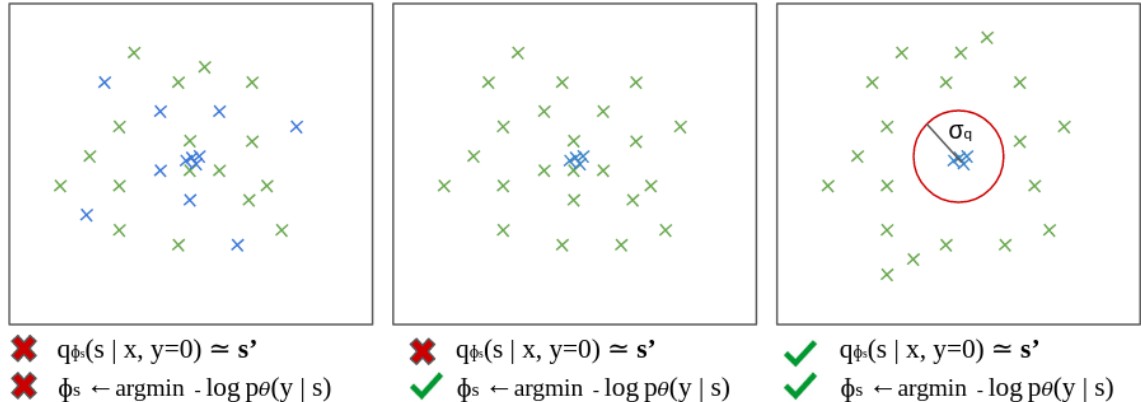

Figure 5: Illustration of the regularization loss within the salient space. As in MM-cVAE, the prior $q_{\phi_s(s|x,y=0)} \sim$ **s'** on the background samples (blue) forces their variance to be as small as possible. However, as the prior on target samples (green) follow a normal distribution, they may overlap with the background distribution. To avoid this case, our method trains a non-linear classifier to avoid the overlap of both distributions with a margin proportional to $\sigma_q$.

## Appendix C. The effect of matching target and background distributions despite data biases

In the paper, we pinpointed data imbalance as a possible example of data biases. We argued that forcing the distribution in the common space to be the same across target and background samples may undermine the capture of common factors in the common space since some of them might be put in the salient latent space by the method.

We have not used a highly imbalanced dataset to show that behavior but the effect of data biases can still be observed in the neuropsychiatric experiments, Tab. 4. Indeed, in these experiments, the age distribution differs between healthy controls and diseased individuals in the schizophrenia disorder dataset, as shown in (Dufumier, 2023) (Table 2.1: SCHIZCONNECT and BSNIP), which can thus be considered as a data bias. As shown in the lower table, matching the healthy and diseased sample distributions in the common space undermines the capture of patterns associated with the age in the common space of MM-cVAE, but not in SepVAE and ConVAE. Thus, as the reconstruction objective requires the input information to be preserved in the latent space, age-related patterns naturally emerge in the salient space, even if they should be in the common latent space.

Table 4: Separation of healthy and schizophrenia-specific variability experiment. CA-VAE methods performance on the prediction of the common variable AGE, using only factors of test images from the target dataset. MAE=Mean Absolute Error.

|  | AGE MAE SALIENT ↑ | AGE MAE COMMON ↓ |
|---|---|---|
| CONVAE | 7.46±0.18 | 6.40±0.26 |
| MM-CVAE | 7.10±0.34 | 6.55±0.18 |
| SEPVAE | **7.98±0.25** | **6.40±0.13** |

## Appendix D. More details on evaluation

**Evaluation details** Here, we evaluate the ability of SepVAE to separate common from target-specific patterns on three medical and one natural (CelebA) imaging datasets.

For quantitative evaluation, we use the fact that the information about attributes, clinical variables, or subtypes (e.g. glasses/hats in CelebA) should be present either in the common or in the salient space. Once the encoders/decoder are trained, we evaluate the quality of the representations in two steps. First, we train a Logistic (resp. Linear) Regression on the estimated salient and common factors of the training set to predict the attribute presence (resp. attribute value). Then, we evaluate the classification/regression model on the salient and common factors estimated from a test set. By evaluating the performance of the model, we can understand whether the information about the attributes/variables/subtype has been put in the common or salient latent space by the method. Furthermore, we report the background (BG) vs target (TG) classification accuracy. To do so, a 2 layers MLPs is independently trained, except for SepVAE, where salient space predictions are directly estimated by the classifier.

In all Tables, for categorical variables, we compute (Balanced) Accuracy scores (=(B-)ACC), or Area-under Curve scores (=AUC) if the target is binary. For continuous variables, we use Mean Average Error (=MAE). Best results are highlighted in bold, second best results are underlined. For CelebA and Pneumonia experiments, mean, and standard deviations are computed on the results of 5 different runs in order to account for model initializations. For neuro-psychiatric experiments, mean and standard deviations are computed using a 5-fold cross-validation evaluation scheme.

First, the variability within the target dataset is assessed by fitting Logistic (or Linear) Regression to evaluate if the model captures the target-specific variability and discards the common variability. In the case where common attributes are available, we assess if the common space captures these attributes in the same fashion.

Qualitatively, the model can be evaluated by looking at the full image reconstruction (common+salient factors) and by fixing the salient factors to $s'$ for target images. Comparing full reconstructions with common-only reconstructions allows the user to interpret the patterns encoded in the salient factors $s$ (see Fig.2 and Fig.3).

## Appendix E. Implementation Details

### E.1. CelebA glasses and hat versus no accessories

We used a train set of 20000 images, (10000 no accessories, 5000 glasses, 5000 hats) and an independent test set of 4000 images (2000 no accessories, 1000 glasses, 1000 hats), and ran the experiment 5 times to account for initialization uncertainty. Images are of size $64 \times 64$, pixel were normalized between 0 and 1. For this experiment, we use a standard encoder architecture composed of 5 convolutions (channels 3, 32, 32, 64, 128, 256), kernel size 4, stride 2, and padding (1, 1, 1, 1, 1). Then, for each mean and standard deviations predicted (common and salient) we used two linear layers going from 256 to hidden size 32 to (common and salient) latent space size 16. The decoder was set symmetrically. We used the same architecture across all the concurrent works we evaluated. We used a common and latent space dimension of 16 each. The learning rate was set to 0.001 with an Adam

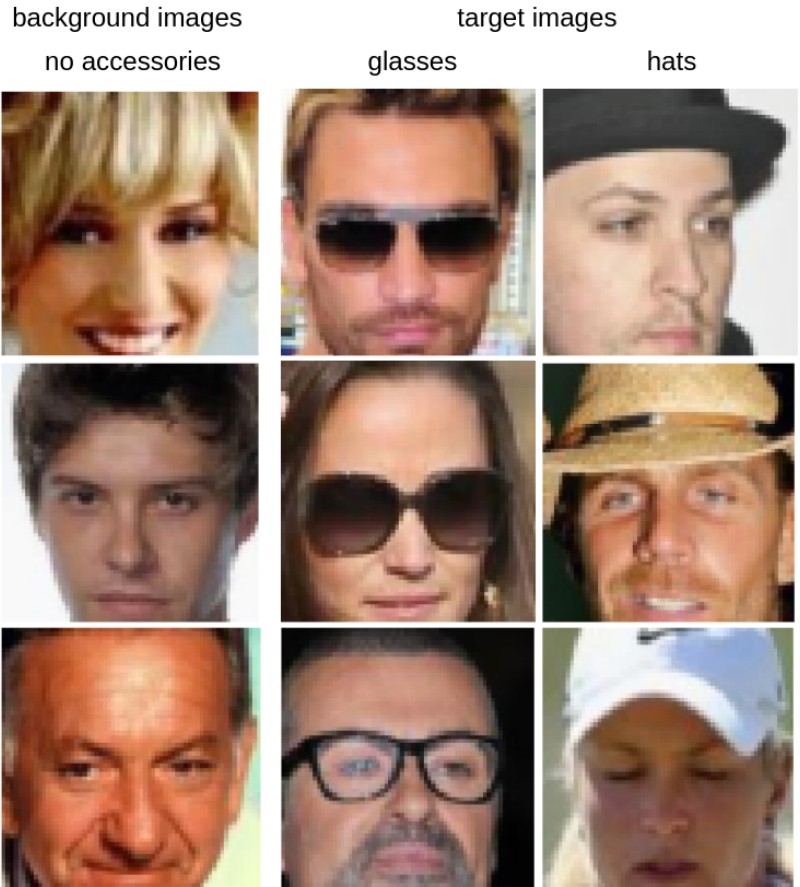

Figure 6: CelebA accessories dataset. We used a train set of 20000 images (10000 no accessories, 5000 glasses, 5000 hats) and an independent test set of 4000 images (2000 no accessories, 1000 glasses, 1000 hats) and ran the experiment 5 times to account for initialization uncertainty. Images were centered on the face and then resized to $64 \times 64$, pixels were normalized between 0 and 1.

optimizer. Oddly we found that re-instantiating it at each epoch led to better results (for concurrent works also), we think that it is because it forgets momentum internal states between the epochs. The models were trained during 250 epochs. To note, in the original contribution, MM-cVAE used latent spaces of 16 for the salient space and 6 for the common space and a different architecture but we noticed that it led to artifacts in the reconstruction (see (Weinberger et al., 2022)). Also, we did not succeed in reproducing their performances with their code, their model, and their latent spaces, even with the same experimental setup. We, therefore, used our model setting which led to better performances across each method with batch size equal to 512. We used $\beta_c = 0.5$ and $\beta_s = 0.5$, $\kappa = 2$, $\gamma = 1e - 10$, $\sigma_p = 0.025$. For MM-cVAE we used the same learning rate, $\beta_c = 0.5$ and $\beta_s = 0.5$, the background salient regularization weight 100, common regularization weight of 1000.

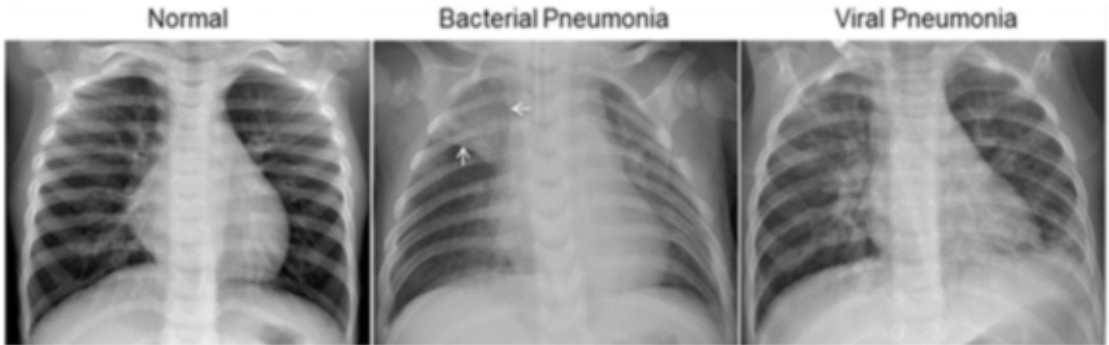

Figure 7: Illustration of the pneumonia dataset. Target images are pneumonia images composed of viral and bacterial pneumonia. Background images are healthy X-Ray images. Original dataset image description from (et al., 2018). The dataset is available at https://www.kaggle.com/datasets/paultimothymooney/chest-xray-pneumonia.

### E.2. Pneumonia

Train set images were graded by 2 radiologist experts and the independent test set was graded by a third expert, the experiment was run 5 times to account for initialization uncertainty. Radiographies were selected from a cohort of pediatric patients aged between one and five years old from Guangzhou Women and Children's Medical Center, Guangzhou. TRAIN set images were graded by 2 radiologists experts and the independent TEST set was graded by a third expert to account for label uncertainty. Images are of size $64 \times 64$, pixel were normalized between 0 and 1. For this experiment, we use a standard encoder architecture composed of 4 convolutions (channels 3, 32, 32, 32, 256), kernel size 4, and padding (1, 1, 1, 0). Then, for each mean and standard deviations predicted (common and salient) we used two linear layers going from 256 to hidden size 256 to (common and salient) latent space size 128. The decoder was set in a symmetrical manner. We used the same architecture across all the concurrent works we evaluated. We used a common and latent space dimension of 128 each. The learning rate was set to 0.001 with an Adam optimizer. Oddly we found that re-instantiating it at each epoch led to better results (for concurrent works also), we think that it is because it forgets momentum internal states between the epochs. The models were trained during 100 epochs with batch size equal to 512. We used $\beta_c = 0.5$ and $\beta_s = 0.1$, $\kappa = 2$, $\gamma = 5e - 10$, $\sigma_p = 0.05$. For MM-cVAE, we used the same learning rate, $\beta_c = 0.5$ and $\beta_s = 0.1$, the background salient regularization weight 100, common regularization weight of 1000.

### E.3. Neuro-psychiatric experiments

The task of identifying consistent correlations between neuro-anatomical biomarkers and observed symptoms in psychiatric diseases is important for developing more precise treatment options. Separating the different latent mechanisms that drive neuro-anatomical variability in psychiatric disorders is a challenging task. Contrastive Analysis (CA) methods such as ours have the potential to identify and separate healthy from pathological neuro-anatomical

patterns in structural MRIs. This ability could be a key component to push forward the understanding of the mechanisms that underlie the development of psychiatric diseases. As explained in the main text, given a background population of Healthy Controls (HC) and a target population suffering from a Mental Disorder (MD), the objective is to capture the pathological factors of variability in the salient space, such as psychiatric and cognitive clinical scores, while isolating the patterns related to demographic variables, such as age and sex, or acquisition sites to the common space. For each experiment, we gather T1w anatomical VBM (Ashburner and Friston, 2000) pre-processed images of HC and MD subjects of size $128 \times 128 \times 128$. We divide them into 5 TRAIN, VAL splits (0.75, 0.25) and evaluate in a cross-validation scheme the performance of SOTA CA-VAEs. Let us note that this is a challenging problem, especially due to the high dimensionality of the input and the scarcity of the data. Notably, the measures of psychiatric and cognitive clinical scores are only available for some patients, making it scarce and precious information.

Images are of size $128 \times 128 \times 128$ with voxels normalized on a Gaussian distribution per image. Experiments were run 3 times with a different train/val/test split to account for initialization and data uncertainty. For this experiment, we use a standard encoder architecture composed of 5 3D-convolutions (channels 1, 32, 64, 128), kernel size 3, stride 2, and padding 1 followed by batch normalization layers. Then, for each mean and standard deviations predicted (common and salient), we used two linear layers going from 32768 to hidden size 2048 to (common and salient) latent space size 128. The decoder was set symmetrically, except that it has four transposed convolutions (channels 128, 64, 32, 16, 1), kernel size 3, stride 2, and padding 1 followed by batch normalization layers. We used the same architecture across all the concurrent works we evaluated. We used a common and latent space dimension of 128 each. The models were trained during 51 epochs with a batch size equal to 32 with an Adam optimizer. For the Schizophrenia experiment, for Sep VAE, we used a learning rate of 0.00005, $\beta_c = 1$ and $\beta_s = 0.1$, $\kappa = 10$, $\gamma = 1e - 8$, $\alpha = \frac{1}{0.01}$. For MM-cVAE we used the same learning rate, $\beta_c = 1$ and $\beta_s = 0.1$, the background salient regularization weight 100, common regularization weight of 1000. For the Autism disorder experiment, we used a learning rate of 0.00002, $\beta_c = 1$ and $\beta_s = 0.1$, $\kappa = 10$, $\gamma = 1e - 8$, $\sigma_p = 0.01$. For MM-cVAE we used the same learning rate, $\beta_c = 1$ and $\beta_s = 0.1$, the background salient regularization weight 100, common regularization weight of 1000.

## Appendix F. On Mutual Information Estimation and Minimization

To promote independence between $c$ and $s$, we minimize their mutual information, defined as the KL divergence between the joint distribution $q(c, s)$ and the product of their marginals $q(c)q(s)$. However, computing this quantity is not trivial, and it requires a few tricks to correctly estimate and minimize it. As in (Abid and Zou, 2019), it is possible to take inspiration from FactorVAE (Kim and Mnih, 2019), which proposes to estimate the density-ratio between a joint distribution and the product of the marginals. In our case, we seek to enforce the independence between two sets of latent variables rather than between each latent variable of a set. The density-ratio trick (Nguyen et al., 2010; Sugiyama et al., 2012) allows us to estimate the quantity inside the log in Eq.5. First, we sample from $q(c, s)$ by randomly choosing a batch of images $(x_i, y_i)$ and drawing their latent factors $[c_i, s_i]$ from the encoders $e_{\phi_c}$ and $e_{\phi_s}$. Then, we sample from $q(c)q(s)$ by using the same batch of

images where we shuffle the latent codes among images (*e.g.*, $[c_1, s_2]$, $[c_2, s_3]$, etc.). Once we obtained samples from both distributions, we trained an **independent** classifier $D_\lambda([c, s])$ to discriminate the samples drawn from the two distributions by minimizing a BCE loss. The classifier is then used to approximate the ratio in the KL divergence, and we can train the encoders $e_{\phi_c}$ and $e_{\phi_s}$ to minimize the resulting loss:

$$\mathcal{L}_{\text{MI}} = \mathbb{E}_{q(c,s)} \log\left(\frac{q(c,s)}{q(c)q(s)}\right) \approx \sum_i \text{ReLU}\left(\log\left(\frac{D_\lambda([c_i, s_i])}{1 - D_\lambda([c_i, s_i])}\right)\right) \tag{5}$$

where the ReLU function forces the estimate of the KL divergence to be positive, thus avoiding to back-propagate wrong estimates of the density ratio due to the simultaneous training of $D_\lambda([c, s])$. Contrarily to (Abid and Zou, 2019), it is important to use an independent optimizer for $D_\lambda$ to ensure that the density ratio is well estimated. The pseudo-code is available in Alg. 1, and a visual explanation is shown in Fig.8.

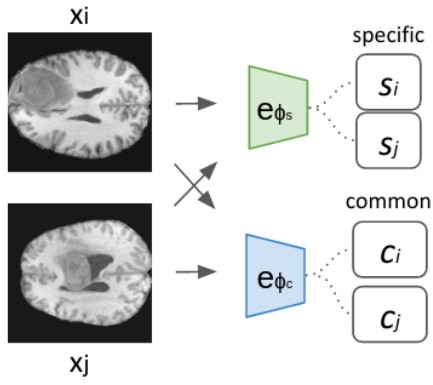

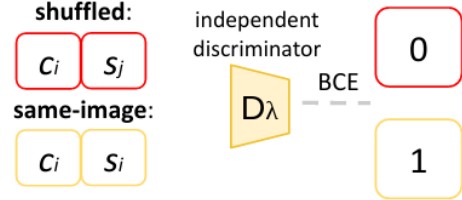

Step 1: train discriminator D independently to predict whether the **common** and **specific** parts are shuffled.

Step 2: train encoders to produce independent latents (that D(.) cannot attribute to 1 or 0) by minimizing the KL.

$$\phi_s, \phi_c \leftarrow \arg\min \phi_s, \phi_c \sum_{i=1}^{N} \text{ReLU}\left(\log\left(\frac{D_\lambda([c,s])}{1 - D_\lambda([c,s])}\right)\right)$$

Figure 8: Illustration of Mutual Information loss between the common and the salient space. Given two images $x_a$ and $x_b$, 4 sets of latents are computed: $c_a$ and $s_a$ latents of the image $a$, $c_b$ and $s_b$ latents of the image $b$. A non-linear MLP is independently trained with a binary cross-entropy loss to classify shuffled concatenations (i.e., from different images) with the label 0 and concatenations of latents coming from the same image with label 1. Then, during training, encoders should not to be able to identify whether a concatenation of latents belong to class 0 (shuffled common and salient spaces) or class 1 (common and salient spaces coming from the same image). We encourage that by minimizing $D_{KL}(p_{\phi_s,\phi_c}(c,s)||p_{\phi_c}(c) \times p_{\phi_s}(s))$.

**Algorithm 1:** Minimizing the Mutual Information between common and salient spaces, given a batch of size $B$.

1: **Input:** $X \in \mathbf{R}^{B \times (C \times W \times H)}$

2: **for** $t$ in epochs : **do**

3:     $\underline{\text{Discriminator training :}}$

4:     Sample $z = [c, s]$ from $q_{\phi_c, \phi_s}$.

5:     Sample $\bar{z} = [c, \bar{s}]$ from $q_{\phi_c} \times q_{\phi_s}$ by shuffling $s$ along the batch dimension.

6:     Compute $\mathcal{L}_{BCE} = -\log(D(z)) - \log(1 - D(\bar{z}))$

7:     Freeze $\phi_c$ and $\phi_s$. Update $D$ parameters only.

8:     $\underline{\text{Encoders training :}}$

9:     Sample $z = [e_{\phi_c}(x), e_{\phi_s}(x)]$ from $q_{\phi_c, \phi_s}$.

10:     Compute $\mathcal{L}_{MI} = \sum_{i=1}^{B} \text{ReLU}\left( \log \frac{D(z_i)}{1 - D(z_i)} \right)$

11:     Freeze $D$ parameters. Update $\phi_c$ and $\phi_s$.

12: **end for**

