# OpenReview forum: "SepVAE: a contrastive VAE to separate pathological patterns from healthy ones"
_MIDL.io/2024/Conference — MIDL 2024 Poster_

### Official Review · Reviewer_D7R9 · 2024-02-27

**Confidence:** 3
**Preliminary Rating:** 4
**Recommendation:** Poster
**Final Rating:** 4

**Summary:**

This work introduces a  Contrastive Analysis method with which we estimate the common generative factors and the ones that are
target-specific. The motivation for his lies in distinguishing variations like aging, gender from markers which are Alzheimer’s specific like temporal lobe atrophy

**Strengths:**

1) I think this paper does a good job in explaining the motivation and how well the method works to the reader. Figures 2 and 3 capsulate the core use case of SepVAE.

2) I believe the main novelty of the work lies in implementing  salient space discriminability and the salient/common independence, and the justification for this seems sound to me.

3) The results look good to me with more detailed analysis in the PCA projections as well which further validate the separability and usefulness of the proposed method.

**Weaknesses:**

While the proposed method definitely has improvements over the previous methods, it is not clear to me how exactly the introduced loss terms in the proposed loss is helpful in the final performance. I understand that Table 2 tries to show this, the authors basically remove each individual loss terms to actually show how much each loss terms influence. I think it might be better if the authors show how adding each individual term actually affects the performance and so we can also understand how much of an overlap is present between the loss in the previous methods ConVAE MM-VAE.

**Detailed Comments:**

NA

**Justification Of Final Rating:**

The reviewers have addressed my comments and I keep my initial rating of accept as the novelty of the work lies in implementing salient space discriminability and the salient/common independence, and SepVAE has also been well experimented and ablated through experiments.

**Justification Of The Preliminary Rating:**

I am giving a Weak Accept for this paper as the paper introduces salient space discriminability and the salient/common independence in the contrastive VAE. However, I also acknowledge that my knowledge is limited in this area as con VAE (2019) has been a well established method and I am not sure if there are recent methods which do something similar.

**Questions To Address In The Rebuttal:**

It would be great to have more details on how exactly the work is different from the previous VAE methods proposed to differentiate. The writing can be changed such that the focus is more on how exactly this method is different from the conVAE based previous methods. I understand that the authors try to do this in the end of Related Works section but I do not feel it is sufficient.

**Special Issue:**

No

---

> ### Author Response · Authors · 2024-03-14
> **Response to RD7R9**
>
> 1) ''I think it might be better if the authors show how adding each individual term actually affects the performance and so we can also understand how much of an overlap is present between the loss in the previous methods ConVAE MM-VAE'' \\
>
> Answer:  As suggested by the Reviewer, we added the missing ablation combinations (MI + SAL, MI + CLSF, and MI + SAL + CLSF) in the Ablation Study of the original paper (Table. 2) and specified which combination refers to the previous method ConVAE.
>
>
> 2) ''It would be great to have more details on how exactly the work is different from the previous VAE methods proposed to differentiate. The writing can be changed such that the focus is more on how exactly this method is different from the conVAE-based previous methods. I understand that the authors try to do this at the end of the Related Works section but I do not feel it is sufficient.''
>
> Answer: We have modified the revised version of the manuscript to better clarify the differences between our method and the previous works. Changes have been highlighted in blue.

---

> > ### Comment · Reviewer_D7R9 · 2024-03-22
> >
> > I thank the reviewers for their comments and believe the papers deserves to be accepted to MIDL.

---

### Official Review · Reviewer_zXYa · 2024-02-28

**Confidence:** 3
**Preliminary Rating:** 4
**Recommendation:** Oral
**Final Rating:** 4

**Summary:**

The authors introduce a novel Contrastive Variation Autoencoder (CA-VAE) model equipped with two new regularization losses: disentangling between common and salient representations, and classification between background and target samples in the salient space. This enhancement addresses limitations in existing CA-VAE models, such as information sharing between latent spaces and incomplete capture of salient factors of variation. The performance of the proposed model is demonstrated on three medical datasets and the CelebA dataset.

**Strengths:**

The authors effectively outline the limitations of current state-of-the-art methods to justify their proposed solution.

Their solution is grounded in theoretical principles, supported by thorough experimental analysis to evaluate its performance.

**Weaknesses:**

The specific term in the loss function responsible for discriminating between subgroups is not clearly identified or explained.

Robustness against data imbalance is not justified

The paper lacks experimental results demonstrating the use of the extracted features to train a competitive diagnostic algorithm, such as a diagnostic classification model.

**Detailed Comments:**

The paper is well-written and coherent, with theoretical justifications provided for the proposed ideas.

One interesting aspect is the discussion on robustness against data imbalance. The authors mention a scenario involving an imbalanced dataset (page 3; Matching background and target common patterns section) but do not provide experimental evidence to support the claim of robustness.

Regarding Equation (3); Shouldn't the term in the denominator of the log be D_lambda([c_i,s_j])

**Justification Of Final Rating:**

I would like to thank the authors for reponding the review comments. The authors, by and large, replied to my comments. After the reviews, my score is more positive but still not at the level of strong accept. Therefore I will keep my score as it is.

**Justification Of The Preliminary Rating:**

The subject topic of the paper; disentangling  salient and common information during feature representation learning, is an important topic  and the authors provide a novel solution to this problem.

Their solution is grounded in theoretical principles, supported by thorough experimental analysis to evaluate its performance.

I only give weak accept, because authors did not provide experiments to show that the extracted features are good for developing a competitive diagnostic algorithm (e.g. diagnostic classification). If they can provide some results towards this end, I am inclined to increase my score to strong accept.

**Questions To Address In The Rebuttal:**

It is unclear from the paper which specific term in the loss function is responsible for discrimination between subgroups. It's uncertain whether this discrimination arises as a by-product of the other losses.

It would be beneficial to provide evidence demonstrating the model's robustness against data imbalance.

The paper lacks clarity on how competitive the extracted features are when applied to a downstream task. Specifically, it is unclear whether these features enable the training of diagnostic models that perform at the level of state-of-the-art algorithms reported in the literature.

**Special Issue:**

Yes

---

> ### Author Response · Authors · 2024-03-14
> **Response to RzXYa's first point**
>
> ''It is unclear from the paper which specific term in the loss function is responsible for discrimination between subgroups. It's uncertain whether this discrimination arises as a by-product of the other losses.''
>
> Answer: The unsupervised discrimination between the subgroups arises from \textit{all} the proposed regularization losses which participate, as shown in the ablation study, to correctly separate the common from the salient information. Indeed, the discriminative information about the subgroups should only be in the salient space and not in the common space. Being an \textit{unsupervised} validation, there is no knowledge or information about the subgroups during training.

---

> ### Author Response · Authors · 2024-03-14
> **Response to RzXYa's second point**
>
> "It would be beneficial to provide evidence demonstrating the model's robustness against data imbalance."
>
> Answer: In the paper, we pinpointed data imbalance as a possible example of data biases. We argued that forcing the distribution in the common space to be the same across target and background samples may undermine the capture of common factors in the common space, since some of them might be actually put in the salient latent space by the method.
>
> We have not used a highly imbalanced dataset to show that behavior but the effect of data biases can still be observed in the neuropsychiatric experiments. Indeed, in these experiments, the age distribution differs between healthy controls and diseased individuals in the schizophrenia disorder dataset, as shown in [A], Table 2.1 (SCHIZCONNECT and BSNIP), which can thus be considered as a data bias. As shown in the lower table,
> matching the healthy and diseased sample distributions in the common space undermines the capture of patterns associated with the age in the common space of MM-cVAE, but not in SepVAE and ConVAE. Thus, as the reconstruction objective requires the input information to be preserved in the latent space (either common or salient), age-related patterns naturally emerge in the salient space, even if they should be in the common latent space. We will add this Table and discussion in the Supplementary of the revised version of the manuscript.
>
> |         | Age MAE S$\uparrow$    | Age MAE C $\downarrow$    |
> |---------|------------------------|---------------------------|
> | ConVAE  | \underline{7.46$\pm$0.18}          | \underline{6.40$\pm$0.26} |
> | MM-cVAE | 7.10$\pm$0.34          | 6.55$\pm$0.18             |
> | SepVAE  | \textbf{7.98$\pm$0.25} | \textbf{6.40$\pm$0.13}    |
>
> [A] B. Dufumier, Representation learning in neuroimaging
> Transferring from big healthy data to small clinical cohorts, Thesis Manuscript.

---

> ### Author Response · Authors · 2024-03-14
> **Response to RzXYa's third point**
>
> ''The paper lacks clarity on how competitive the extracted features are when applied to a downstream task. Specifically, it is unclear whether these features enable the training of diagnostic models that perform at the level of state-of-the-art algorithms reported in the literature.''
>
>
> Answer: First of all, we can observe that we reported the diagnosis classification scores in the salient space and the common space in all our experiments ("BG vs TG").
>
> The salient latent space reaches an AUC of around $79.15\pm3.39$ \% in discriminating schizophrenia individuals from healthy controls, which is close to what has been reported in the literature (from 79\% to 81\% in [A]) with deeper convolutional architectures. Similarly, the salient latent space reaches an AUC of around $59.73\pm1.78$ \% in discriminating autistic individuals from healthy controls, which is close to what has been reported in the literature (from 59.5\% to 61\% in [A]) with deeper convolutional architectures.
>
> On BRATS-2021, SepVAE reaches a balanced accuracy of $99.31$ \%, which is close to what has been reported in the literature (as [B] reported a classification result of $99.98$\% with a pre-trained Inception-ResnetV2 neural network, and [C] reported a classification result of $99.29$).
>
> [A] B. Dufumier, Representation learning in neuroimaging
> Transferring from big healthy data to small clinical cohorts, Thesis Manuscript.
>
> [B] D. S. Islam, R. Udhayakumar, S. Kar, P. N, U. S. Aswal and D. K. J. B. Saini, "Enhancing Brain Tumor Detection Classification by Computational Intelligence," 2023 Second International Conference on Electronics and Renewable Systems (ICEARS), Tuticorin, India, 2023, pp. 1044-1049.
>
> [C] G. Bompem, D. Pandluri, Batch Normalization Based Convolutional Neural Network for Segmentation and Classification of Brain Tumor MRI Images, INASS 2023.

---

### Official Review · Reviewer_8ZKa · 2024-03-01

**Confidence:** 4
**Preliminary Rating:** 4
**Recommendation:** Poster

**Summary:**

Paper Summary: The paper presents SepVAE, a novel approach within the family of Contrastive Analysis Variational Auto-Encoders (CA-VAEs), which aims to distinguish between common factors of variation in a background dataset (typically representing healthy subjects) and unique factors in a target dataset (typically representing patients). The main innovation of SepVAE lies in its introduction of two new regularization losses to improve the separation between common and salient (target-specific) features in the latent space: a disentangling term that differentiates between common and salient representations, and a classification term that discriminates between background and target samples in the salient space. According to the authors, SepVAE outperforms existing CA-VAE models in separating pathological patterns from healthy ones across three medical applications and a natural image dataset (CelebA).

Strength: SepVAE introduces novel regularization losses to better separate common and salient features between datasets, addressing a significant limitation in existing CA-VAEs. The performance is also outstanding comparing to existing baselines. The paper is in general well-organized with no English grammar errors.

Weakness: While the paper demonstrates improvements over existing methods, the evaluation is somewhat limited to specific types of medical data and the CelebA dataset, which might not cover all potential use cases or data types. Will the authors consider implementing the model on other medical image dataset? Another suggestion is that, an ablation study may be helpful here. The authors may consider removing some structures in the SepVAE model (such as the novel regularization losses) and compare the corresponding results. In this case people can better understand where the efficiency of this model comes from.

In conclusion, this is a typical application of deep learning techniques to medical imaging. Although there are some weakness as mentioned above, this is a fairly successful paper. So, I will suggest to accept this paper after minor revision. The authors may want to address the weakness I mentioned in above.

**Strengths:**

Strength: SepVAE introduces novel regularization losses to better separate common and salient features between datasets, addressing a significant limitation in existing CA-VAEs. The performance is also outstanding comparing to existing baselines. The paper is in general well-organized with no English grammar errors.

**Weaknesses:**

Weakness: While the paper demonstrates improvements over existing methods, the evaluation is somewhat limited to specific types of medical data and the CelebA dataset, which might not cover all potential use cases or data types. Will the authors consider implementing the model on other medical image dataset? Another suggestion is that, an ablation study may be helpful here. The authors may consider removing some structures in the SepVAE model (such as the novel regularization losses) and compare the corresponding results. In this case people can better understand where the efficiency of this model comes from.

**Detailed Comments:**

Paper Summary: The paper presents SepVAE, a novel approach within the family of Contrastive Analysis Variational Auto-Encoders (CA-VAEs), which aims to distinguish between common factors of variation in a background dataset (typically representing healthy subjects) and unique factors in a target dataset (typically representing patients). The main innovation of SepVAE lies in its introduction of two new regularization losses to improve the separation between common and salient (target-specific) features in the latent space: a disentangling term that differentiates between common and salient representations, and a classification term that discriminates between background and target samples in the salient space. According to the authors, SepVAE outperforms existing CA-VAE models in separating pathological patterns from healthy ones across three medical applications and a natural image dataset (CelebA).

Strength: SepVAE introduces novel regularization losses to better separate common and salient features between datasets, addressing a significant limitation in existing CA-VAEs. The performance is also outstanding comparing to existing baselines. The paper is in general well-organized with no English grammar errors.

Weakness: While the paper demonstrates improvements over existing methods, the evaluation is somewhat limited to specific types of medical data and the CelebA dataset, which might not cover all potential use cases or data types. Will the authors consider implementing the model on other medical image dataset? Another suggestion is that, an ablation study may be helpful here. The authors may consider removing some structures in the SepVAE model (such as the novel regularization losses) and compare the corresponding results. In this case people can better understand where the efficiency of this model comes from.

In conclusion, this is a typical application of deep learning techniques to medical imaging. Although there are some weakness as mentioned above, this is a fairly successful paper. So, I will suggest to accept this paper after minor revision. The authors may want to address the weakness I mentioned in above.

**Justification Of The Preliminary Rating:**

In conclusion, this is a typical application of deep learning techniques to medical imaging. Although there are some weakness as mentioned above, this is a fairly successful paper. So, I will suggest to accept this paper after minor revision. The authors may want to address the weakness I mentioned in above.

**Questions To Address In The Rebuttal:**

Not applicable.

**Special Issue:**

No

---

> ### Author Response · Authors · 2024-03-14
> **Response to R8ZKa's first point.**
>
> ''The evaluation is somewhat limited to specific types of medical data and the CelebA dataset, which might not cover all potential use cases or data types. Will the authors consider implementing the model on other medical image datasets?''
>
> This is an interesting point. In this article, we have focused on \textit{anatomical} imaging data (T1w MRI and X-Ray) providing quantitative and qualitative results on three medical applications and one natural images dataset (CelebA). We agree with the Reviewer that extending the proposed method to other anatomical imaging modalities, such as diffusion MRI or histopathology, or even to functional imaging data, like fMRI or EEG, would be of interest and we are currently working on it. Another interesting extension would be the integration of non-imaging modalities (e.g., clinical, genetic data). Even if the methodological framework is rather generic and could be easily extended to such new modalities, the encoder-decoder architecture should be probably changed and adapted to the modality under analysis. This is left as future work.

---

> ### Author Response · Authors · 2024-03-14
> **Response to R8ZKa's second point.**
>
> ''Another suggestion is that an ablation study may be helpful here. The authors may consider removing some structures in the SepVAE model (such as the novel regularization losses) and comparing the corresponding results so that people can better understand where the efficiency of this model comes from.''
>
>
> Answer:  We provided an ablation study in Table. 2 in the original manuscript. As suggested by Reviewer D7R9, we enrich it by providing the three remaining loss combinations (MI + SAL, MI + CLSF, and MI + SAL + CLSF). We report the new results in the following table. Please note that ConVAE is equivalent to SepVAE without the CLSF and the SAL losses. We clarify that in the revised version of the manuscript writing "\textit{no SAL means that we ignored the informationless constraint on the Salient Prior loss for background samples (as in Abid and Zou)}."
>
> |                                           | Subgrp Acc                 | Subgrp Acc                 | Bg vs Tg Acc            | Bg vs Tg Acc               |
> |-------------------------------------------|----------------------------|----------------------------|-------------------------|----------------------------|
> |                                           | salient $\uparrow$         | common $\downarrow$        | salient $\uparrow$      | common $\downarrow$        |
> | ConVAE (:= SepVAE no \textbf{CLSF + SAL}) | 82.30$\pm$1.53             | 73.58$\pm$1.84             | 67.80$\pm$5.93          | 58.05$\pm$7.17             |
> | MM-cVAE                                   | 82.86$\pm$1.87             | 74.35$\pm$3.19             | 70.44$\pm$2.69          | 59.94$\pm$5.88             |
> | SepVAE                                    | \textbf{84.78$\pm$0.42}    | \textbf{70.92$\pm$1.39}    | \textbf{78.13$\pm$3.03} | 57.52$\pm$4.14             |
> | SepVAE no \textbf{MI}                     | 84.10$\pm$0.48             | 71.79$\pm$2.94             | 75.18$\pm$5.69          | 60.35$\pm$4.73             |
> | SepVAE no \textbf{CLSF}                   | \underline{84.71$\pm$1.19} | 73.58$\pm$2.19             | 71.91$\pm$4.65          | \underline{55.79$\pm$5.41} |
> | SepVAE no \textbf{SAL}                    | 83.98$\pm$0.85             | 72.61$\pm$2.05             | 73.03$\pm$2.97          | 61.43$\pm$2.25             |
> | SepVAE no \textbf{MI + SAL}               | 81.58$\pm$3.68             | \underline{71.73$\pm$5.17} | 61.24$\pm$3.89          | \textbf{54.33$\pm$5.30}    |
> | SepVAE no \textbf{MI + CLSF}              | 84.25$\pm$0.47             | 73.17$\pm$3.15             | 53.10$\pm$1.63          | 57.58$\pm$6.74             |
> | SepVAE no \textbf{MI + SAL + CLSF}        | 81.78$\pm$2.12             | 76.71$\pm$2.10             | 62.87$\pm$7.15          | 59.37$\pm$5.69             |

---

### Author Response · Authors · 2024-03-14

We thank the Reviewers for their thoughtful and useful feedback. We have provided our response to each of the points raised. Based on the received reviews, we have updated our manuscript (changes highlighted in blue).

---

### Meta-Review · Area_Chair_PNvr · 2024-04-04

**Recommendation:** Accept (Poster)
**Confidence:** 4

**Metareview:**

The paper introduces SepVAE, a Contrastive Analysis Variational Auto-Encoder (CA-VAE) variant designed to distinguish between common and salient features in datasets, particularly useful in medical imaging. SepVAE introduces new regularization losses to enhance this separation, improving upon existing CA-VAE models. The performance is evaluated on medical datasets and CelebA, demonstrating superiority over baselines. Reviewers generally praise the paper's organization, innovation, and performance but highlight some limitations and suggest improvements.

Despite some limitations, the reviewers generally recommend accepting the paper with minor revisions. Suggestions include expanding the evaluation to other medical datasets, conducting ablation studies to clarify the impact of the new loss terms, and including experiments with diagnostic algorithms to demonstrate practical utility. Addressing these suggestions would enhance the paper's contribution and make it more compelling for publication.

---

### Decision · Program_Chairs · 2024-04-06

Accept (Poster)